# Relative Validity of the Groningen IBD Nutritional Questionnaire (GINQ-FFQ): A Food Frequency Questionnaire Designed to Assess Nutritional Intake in Patients with Inflammatory Bowel Disease

**DOI:** 10.3390/nu17020239

**Published:** 2025-01-10

**Authors:** Iris Barth, Corien L. Stevens, Vera Peters, Desiree A. Lucassen, Edith J. M. Feskens, Gerard Dijkstra, Marjo J. E. Campmans-Kuijpers

**Affiliations:** 1Department of Gastroenterology and Hepatology, University Medical Centre Groningen (UMCG), 9713 GZ Groningen, The Netherlandsgerard.dijkstra@umcg.nl (G.D.); m.j.e.campmans-kuijpers@umcg.nl (M.J.E.C.-K.); 2Graduate School of Medical Sciences (GSMS), University of Groningen (RUG), 9700 AB Groningen, The Netherlands; 3Division of Human Nutrition and Health, Wageningen University & Research, 6700 AB Wageningen, The Netherlands; desiree.lucassen@wur.nl (D.A.L.); edith.feskens@wur.nl (E.J.M.F.)

**Keywords:** inflammatory bowel disease [IBD], food frequency questionnaire [FFQ], validation, nutritional assessment

## Abstract

**Background and Objective:** To assess nutritional intake of patients with inflammatory bowel disease (IBD), a disease-specific food frequency questionnaire (FFQ) was developed: the Groningen IBD Nutritional Questionnaire (GINQ-FFQ). Aim of this study was to assess the relative validity of the GINQ-FFQ. **Methods:** Between 2019 and 2022, participants of the 1000IBD cohort were included and filled out a 3-day food diary and the GINQ-FFQ. Nutritional intake of nutrients and food groups was calculated. Bland–Altman analysis was conducted for energy intake, while paired *t*-tests and Wilcoxon signed rank tests were used for nutrient and food group intake. Additionally, group-level bias, cross-classification, and correlation analysis were performed. **Results:** 142 patients (59.2% females, mean age of 49 ± 14 years) were included. Bland-Altman analysis showed a mean difference between the GINQ-FFQ and 3FD of –63.6 kcal (±638.4), with limits of agreement ranging from –1315 to 1188 kcal. Differences in energy intake was significantly associated with higher mean total energy intake (*p* < 0.001). When stratifying for sex, this association only was significant for males. Group-level bias showed that the GINQ-FFQ tends to result in lower intake reports for macro- and micronutrients. Ranking ability (cross-classification) of macro-, micronutrients and food groups was good. Correlation coefficients for nutrients and food groups were considered acceptable or good. **Conclusions:** Overall, the GINQ-FFQ is a valid food frequency questionnaire to assess nutritional intake specifically for patients with IBD. However, for males with high total energy intakes, dietary assessment could be less accurate.

## 1. Introduction

Inflammatory bowel disease (IBD) is an autoimmune disease comprising both Crohn’s disease (CD) and ulcerative colitis (UC) [1]. Studies have shown that diet plays an important role in the development and course of IBD [2,3]. This specific patient population is especially vulnerable to malnutrition because of pathophysiological changes caused by the disease, such as increased nutrient demand due to inflammation as well as anorexia caused by symptoms of the disease [4]. Patients often start to experiment with the composition of their diet, since many patients can directly link their gastrointestinal symptoms to their intake [5,6]. When leaving out certain food items or even entire food groups without proper dietary guidance, risk for malnutrition increases even more [4].

To be able to provide proper dietary guidance, professionals are in need of valid dietary assessment tools specifically designed and validated for the IBD population. These assessment tools will allow researchers to register dietary intake of patients with IBD correctly by adequately identifying disease-specific deficiencies in nutritional intake. This will lead to more robust nutritional studies that can be used to develop new evidence-based dietary guidelines for the IBD population. Aside from benefits for research, these tools could also be used in clinical evaluation of patients and thus directly assist health care providers in formulating specific dietary advice [7].

In 2019, in collaboration with Wageningen University, division Human Nutrition and Health, our research group developed the Groningen IBD Nutritional Questionnaires (GINQ-FFQ), a food frequency questionnaire designed to assess dietary intake in patients with IBD. Development, face, and content validity of the GINQ-FFQ are described elsewhere [7]. Validation of dietary intake assessment tools must take place to show the magnitude and direction of possible measurement error. Validation can also be used to investigate potential causes for these measurement errors and consequently identify ways to minimize or account for these errors in future analyses [8]. Therefore, the aim of this study is to assess the relative validity of the GINQ-FFQ compared with a 3-day food diary (3FD).

## 2. Materials and Methods

### 2.1. Patients and Dietary Assessment

All included patients were part of the 1000IBD cohort [9]. This cohort is part of the larger “Parelsnoer” Initiative (PSI) [10,11]. The PSI was established to optimize clinical biobanking within the eight Dutch university medical centres, specifically for research purposes. The PSI was approved by the medical research ethics committee of the University Medical Centre Groningen (METC UMCG 2008.338, 5 August 2009). This validation cohort study, within the 1000IBD project, was approved by the medical research ethics committee of the University Medical Centre Groningen (METC UMCG 2019.451, 27 August 2019). All participants provided digital informed consent.

Between October 2019 and August 2022, all participants who were approached to participate in the sampling round of the 1000IBD cohort were also invited to participate in this validation study. Participants were asked to fill out a 3FD on paper, which was collected together with the samples for the 1000IBD cohort. At the same time, participants received a weblink and personal code to fill out the online GINQ-FFQ. Apart from the GINQ-FFQ, participants additionally filled out questionnaires in Dutch to assess baseline characteristics of the 1000IBD cohort through the REDCap data capturing tool. These included questionnaires to evaluate disease activity (Monitor IBD At Home questionnaire (MIAH)) [12], food-related quality of life (Fr-QoL-29) [13], physical activity (Baecke questionnaire) [14], and some questions to assess socio-economic status and educational level. Circa 378 patients from the University Medical Centre Groningen were invited to participate in this study. Overall, there were 270 responses, of which 116 were incomplete. Of the remaining 154 participants, 12 were excluded from analysis due to implausible energy intake. Over- and underreporting of daily energy intake (kcal) was determined for both the GINQ-FFQ and 3FD using cut-off values of <500 and >3500 kcal for women and <800 and >4000 for men [15,16]. The final analyses included 142 participants (Figure 1). There is no clear consensus on the appropriate sample size for validation studies of nutritional questionnaires. Cade et al. reviewed 227 FFQ validation studies and found a median sample size of 255 (range 6–3750), advising the use of at least 50–100 subjects per demographic group [17]. Similarly, Willet et al. suggested that for correlations between 0.5 and 0.7 between questionnaires, sample sizes of 100–200 participants are sufficient [18]. These recommendations were supported by two more recent validation studies [19,20]. Therefore, we concluded that a sample size of 50–200 was sufficient for our validation study.

### 2.2. Calculation of Nutritional Intake

The GINQ-FFQ was developed to assess dietary intake, specifically in patients with IBD, over the previous month. This FFQ contains 121 food frequency questions assessing 218 food items [7]. To fill out the GINQ-FFQ, participants received a personal link to a secured website, The Dutch FFQ-Tool^TM^, developed by Wageningen University and Research (WUR). The FFQ-Tool^TM^ uses ranges of standard household measures and common Dutch portion sizes and frequency questions with specific weightings to determine the daily amounts used on average over the past month [21]. To convert the daily intake amounts into nutrient intake, the FFQ-Tool^TM^ was linked to the Dutch food composition database (NEVO, version 2010/2.0) [22].

For the 3FD, patients were asked to write down their nutritional intake on one weekend day and two non-consecutive weekdays. The collection of this data was fully self-reported, without direct supervision of a healthcare provider. In the 3FD, individual portion sizes of food consumption were taken into account as much as possible by providing examples of household measures. Based on these measures, patients could indicate how many millilitres their glass, cup/mug, bowl, deep plate, wine, or beer glass contained, used for food consumption. Supplement 2 shows the English translation of the used 3FD with all the instruction participants received. After finalizing all data collection, nutritional intake of the 3FD was calculated by dietitians using a standardized protocol developed for this study. For this purpose, a Dutch nutritional calculation tool was used (Evry, version 2.3.7.0 [23]), which converts food consumption into nutrient intake using the same NEVO database as applied to the GINQ-FFQ. The checking of the 3FD and any correction thereof was carried out by the researchers. Total daily nutrient intake from the 3 days in the 3FD was calculated by multiplying the portion size with 0.33 (1/3) [24]. For both 3FD and GINQ-FFQ, nutrient intake was assessed and was part of the overall dietary intake calculation.

Furthermore, for both the 3FD and GINQ-FFQ, we determined the energetic contribution (En%) of carbohydrates, protein, and fat using the following equations:En% carbohydrate = (carbohydrate [g] × 4 kcal)/(total energy [kcal] × 100).En% protein = (protein [g] × 4 kcal)/(total energy [kcal] × 100).En% fat = (fat [g] × 9 kcal)/(total energy [kcal] × 100).

Protein intake per kilogram bodyweight (g/kg) was calculated as follows:Protein intake per kilogram bodyweight = intake [g]/bodyweight [kg].

### 2.3. Food Groups

Food intake derived from the GINQ-FFQ and the 3FDs was categorized into 21 food groups as follows: breakfast products, breakfast grains, dairy, bread and bread substitutes, fats/oils, spreads, eggs, fruits, nuts/stone fruits/seeds, meat and meat substitutes, fish, vegetables, legumes, grain products, sauces, additions to meals, fast food, soup, savoury snacks, sweet snacks, and beverages. These food groups were based on the initial IBD-specific list of food items that was formed by our group when developing the GINQ-FFQ (Appendix A) [7].

### 2.4. Statistics

Data on descriptive statistics were reported based on visual inspection of the normality distribution as follows: mean with standard deviation, median with interquartile range for continuous variables, and number with frequencies for count data. Statistical analyses were performed for both nutrient and food-group intake separately.

To assess systematic differences between the GINQ-FFQ and 3FD and the extent to which the two methods agree, a Bland–Altman analysis was performed with calculation of the mean difference and limits of agreement (1.96 × standard deviation (SD)) [25]. The difference between the energy intake of the GINQ-FFQ and 3FD was plotted against the total mean intake of both methods:Difference in kcal intake (diff kcal) = kcal GINQ − kcal 3FD.Total mean intake = [kcal GINQ + kcal 3FD]/2.Lower limit of agreement = [mean diff kcal] − 1.96 × [SD mean diff kcal].Upper limit of agreement = [mean diff kcal] + 1.96 × [SD mean diff kcal].

For normally distributed nutritional intake data, a paired *t*-test with mean difference was used to further assess the agreement between the two methods. Group-level bias was determined by calculating the ratio of nutrient intake from the GINQ-FFQ to the intake from the 3FD, multiplying the result by 100, and then subtracting 100 to centre the results around zero [26,27].

Positive values indicate an overestimation of the GINQ-FFQ, while negative values are an indication of underestimation compared to the 3FD. A difference of ≥10% was considered an indication of bias [28]. The difference in nutritional intake data between the GINQ-FFQ and 3FD, with non-parametric distributions, was tested with a Wilcoxon signed rank test. A two-tailed *p* value of >0.05 was considered significant [28].

Cross-classification was used to categorize the nutritional intake observations from individuals using the two methods into quintiles, each representing 20% of the total observations. To assess whether these observations were consistent in ranking ability, we determined the percentage of observations that were ranked in the same, adjacent, or extreme quintile. A value of ≥10% in the extreme quintile was considered a poor outcome [28].

To measure the strength and direction of a relationship between the two methods, correlation analysis was performed by using Pearson’s r (*r*). Spearman rank correlation (*rho*) was calculated if the assumption of linearity and normality was violated. Correlation values of ≥0.50 were considered good, 0.21–0.49 acceptable, and ≤0.20 a poor correlation [29].

Since individuals can systematically differ in determinants that influence nutritional intake, total energy intake is generally used as a proxy to account for confounding in nutrition epidemiology studies when investigating causal effects of dietary components on disease outcomes. Therefore, we conducted the aforementioned correlation analyses to address possible confounding by energy intake using the residual method to obtain adjusted correlation coefficients [30]. Log-transformed values were used to address skewed data. Statistical analyses were performed using IBM SPSS Statistics (Version 28) [31] and R Statistical Software (v4.1.2) [32].

## 3. Results

### 3.1. Patient Characteristics

A total of 142 individuals were included in the analyses. Table 1 shows the demographic, disease, and nutritional characteristics of included patients. Mean age was 49 years (SD ± 14 years), where 59.2% were female and 55.6% were patients with CD. A total of 91 (64.1%) patients used nutritional supplements, and 41 (28.9%) reported specific diet prescriptions, dietary habits, or food preferences (e.g., gluten-free diet, lactose-free, or vegetarian).

### 3.2. Energy Intake

Agreement for energy intake between the GINQ-FFQ and 3FD, stratified by sex, is presented in Figure 2 using a Bland–Altman plot. The mean difference between the GINQ-FFQ and 3FD is –63.6 kcal (SD ± 638.4), with 95% limits of agreement (LoA) ranging from –1315 to 1188 kcal. Linear regression analysis demonstrates an association between the GINQ-FFQ and 3FD, with the differences in kcal intake significantly associated with higher mean total energy intake (intercept [α]: −788.7; slope [β]: 0.342 per kcal increase; *p* < 0.001).

For males, the mean difference between the GINQ-FFQ and 3FD is 47.3 kcal (SD ± 780.4), with 95% LoA ranging from −1482 to 1577 kcal (Figure 3). Linear regression analysis demonstrated differences in kcal intake to be significantly associated with higher mean total energy intake (intercept [α]: −955.4; slope [β]: 0.425 per kcal increase; *p* = 0.021).

In females, the mean difference between the GINQ-FFQ and 3FD is −140.2 kcal (SD ± 509.4), with 95% LoA ranging from −1139 to 858.2 kcal (Figure 4). Here, linear regression analysis did not demonstrate a significant association between differences in kcal intake and higher mean total energy intake (intercept [α]: −501.5; slope [β]: 0.184 per kcal increase; *p* = 0.181).

### 3.3. Macro- and Micronutrient Intake

Results of the analyses performed on the absolute nutrient intake per day from both the GINQ-FFQ and 3FD are presented in Table 2.

#### 3.3.1. Macronutrients

When observing mean differences in macronutrient calculations from the GINQ-FFQ and 3FD, the GINQ-FFQ seems to assess smaller means. Paired *t*-tests show significant differences in protein (specifically total protein [g/d] *p* = 0.004 and animal protein [g/d] *p* = 0.002) and fat (specifically total fat [en%] *p* < 0.001 and MUFA [g/d] *p* = 0.021) calculations.

Group-level bias shows a tendency of lower reporting of macronutrients in the GINQ-FFQ compared to the 3FD. Lower assessment is highest for animal protein [g/d] (−11.48%).

Cross-classification shows observations ranked into the extreme quintiles varied from 0.70 to 4.23%, which is considered a good ranking ability.

Total carbohydrates [g/d and En%], mono-/disaccharides [g/d], saturated fat [g/d], cholesterol [mg/d], and alcohol [g/d] show good adjusted correlation coefficients (r > 0.5). Other macronutrients show acceptable correlation coefficients varying between 0.23 and 0.49. When adjusting for energy intake, correlation coefficients changed mildly, but findings of good and poor correlation remain the same.

#### 3.3.2. Micronutrients

There is a significant difference in mean assessment of riboflavin (*p* = 0.004), ascorbic acid (*p* = 0.001), and potassium (*p* = 0.001) between GINQ-FFQ and 3FD.

Group-level bias of micronutrients demonstrates that the majority is <10%; however, the GINQ-FFQ tends to show lower amounts for micronutrient intake. The difference in assessment is most evident for thiamine (−20.10%), riboflavin (−15.33%), and ascorbic acid (−22.50%).

Cross-classification shows overall good ranking ability, as the majority of the nutrients are classified in the same and adjacent quintiles. A significant proportion of observations fall within the extreme quintiles for retinol activity equivalents (6.34%) and cobalamin (5.63%).

Correlation coefficients for micronutrients are overall poor to acceptable, varying from 0.01 to 0.46. Correction for energy intake changes correlation coefficients, but they remain acceptable in approximately half of the micronutrient variables.

#### 3.3.3. Food Group Intake

Table 3 presents the analysis of the food-group intake in grams per day for both the GINQ-FFQ and the 3FD, together with a comparison of the median intake, cross-classification, and correlation coefficients.

Median assessment of food group intake shows a significant difference for 13 out of 21 food groups. Among these food groups, the calculated intake from breakfast products, fats, eggs, nuts, fish, legumes, grains, and savoury was significantly higher in the GINQ-FFQ compared to the 3FD. The intake from other food groups was in general lower in the GINQ-FFQ compared to the 3FD.

Due to the skewed distribution of several food groups, cross-classification was not possible for all 21 groups. For the 12 groups that were cross-classified, the percentage of observations ranked into the extreme quintiles varied from 0.00% to 3.52%. This is considered good ranking ability.

Correlation coefficients between the two assessment methods are overall considered acceptable and good, varying between 0.23 and 0.76. Specifically, the correlation for dairy, bread, and eggs is good (*rho* = 0.57, 0.60, and 0.54, respectively), whereas the correlation for legumes is poor (*rho* = 0.20). After adjusting for energy intake, correlation coefficients demonstrate a change, indicating a poor correlation for sauces and soup (*rho* = 0.16 and 0.17, respectively).

## 4. Discussion

This study aimed to assess the relative validity of the Groningen IBD Nutritional Questionnaire (GINQ-FFQ) compared with a 3-day food diary (3FD). Overall, the GINQ-FFQ is deemed a valid food frequency questionnaire to assess nutritional intake specifically for patients with IBD.

Agreement for total energy intake between GINQ-FFQ and 3FD, assessed by Bland–Altman plot, was good. Linear regression showed that the difference in energy intake between both assessment methods became significantly larger with an increase in mean total energy intake. When stratifying for sex, this association was significant for males, but not for females.

For macronutrients, micronutrients, and food groups, differences in means/medians, group-level bias, cross-classification, and correlation coefficients were assessed. Means for macro- and micronutrient intake were lower in the GINQ-FFQ, differences being significant for total and animal protein, total fat, MUFAs, riboflavin, ascorbic acid, and potassium. For food group intake, the difference in medians was significant in 13 out of 21 food groups. For 8 food groups (breakfast products, fats, eggs, nuts, fish, legumes, grain, and savoury), medians were significantly higher using the GINQ-FFQ assessment, and for 5 food groups (dairy, vegetables, additions, fast food, and beverages), they were significantly lower. Group-level bias assessment showed that the GINQ-FFQ tends to result in lower intake reports for macro- and micronutrients. Cross-classification showed an overall good ranking ability for macro- and micronutrients. Due to skewed distribution, only 12 food groups could be ranked; for these 12 food groups, ranking ability was good. Correlation coefficients for macronutrients and food groups were considered acceptable or good. For micronutrients, approximately half of the correlation coefficients were acceptable. Correlation coefficients for vitamins B1, B2, B6, B11, C, D, and E and zinc were poor.

### 4.1. Complexity of the Validation of a FFQ

In conducting a validation study, food-frequency questionnaire measures are compared with an alternative, but not necessarily more accurate, method of assessing diet. Various methods for validating food frequency questionnaires are reported in the literature [28]. There is no consensus on which type of analysis or combination of analyses should be performed, resulting in inconsistently applied methods. Most of the food frequency validation studies employ three different types of tests, with correlation analysis being the most frequently used [28]. Cut-off values for correlation analysis and for determination of the magnitude of bias in cross-classification and group-level bias may differ between studies and need to be interpreted with caution. Nevertheless, in this validation study, we applied five commonly used tests to conduct an appropriate validation study that fits the GINQ-FFQ [28].

### 4.2. GINQ-FFQ Compared to Other Dutch FFQs

To the knowledge of our research group, the GINQ-FFQ is the first IBD-specific FFQ [7]. A direct comparison of the validity of intake reporting in this population cannot be made. However, there are several Dutch FFQs that are similarly constructed. The Leiden longevity FFQ, the FFQ-NL 1.0, and the Flower FFQ were all built by the same FFQ tool from Wageningen University as the GINQ-FFQ and linked to the same Dutch food composition NEVO database. All 4 FFQs were developed for healthy, Dutch, adult populations. The Leiden Longevity FFQ was used as a base model for the GINQ-FFQ. Reference methods of these validation studies included 24h recalls or the Leiden longevity FFQ, which differ from the 3FD used as a reference method in the present study [7,26,27,33].

Similar to the GINQ-FFQ, the Leiden Longevity FFQ shows increasing differences between the reference method and FFQ with increasing total energy intake [33].

Correlation coefficients for the GINQ-FFQ are within comparable ranges compared to the Leiden Longevity FFQ. Important to note is that the Leiden Longevity used Pearson’s correlation, whereas the present study uses Spearman correlation due to the distribution of our data. Likewise, these results were observed when comparing the correlations between the GINQ-FFQ and the FFQ NL 1.0 [26].

Overall, values of group-level bias in nutrient reporting tend to be higher in the GINQ-FFQ compared to the FFQ-NL 1.0 [26]. The GINQ-FFQ tends to show lower values for group-level bias for nutrient intake compared to the Flower FFQ [27]. However, differences are small.

Although there are methodological and population differences compared to other validation studies of Dutch FFQ’s, overall results are similar, concluding that relative validation was acceptable.

### 4.3. Representativeness of GINQ-FFQ Compared to 3FD

Overall, we observed that the GINQ-FFQ tends to report lower intake of nutrients compared to the 3-day food diary (3FD). This also applies to food group intake, with the exception of eight food groups: breakfast products, fats/oils, eggs, nuts, fish, legumes, grain, and savoury foods. Of these, grains, fats/oils, and legumes are regularly consumed by the Dutch population [34]. It is well established that FFQs, including the GINQ-FFQ, are designed to capture average intake over a longer period, thereby accounting for day-to-day variation in diet [18]. In IBD patients, it is known that nutritional intake varies not only by personal preference or gastrointestinal symptoms but also according to disease activity (active versus remission) [35]. For this reason, the GINQ-FFQ is especially valuable in identifying food groups that are often reduced or excluded from the diets of IBD patients. For instance, median intakes of legumes, fish, eggs, and breakfast products often register as zero in the 3FD but are detected by the GINQ-FFQ [34]. Therefore, the GINQ-FFQ captures more information on episodically consumed food groups compared to the 3FD.

The time needed to fill out an FFQ depends on the number of food items included. Willet et al. reported that approximately 130 food items may be the limit for individuals willing to complete a questionnaire [36]. A FFQ with a larger number of food items takes longer to complete, which could lead to inaccuracy due to fatigue, disinterest, and impaired inattentiveness [36]. The GINQ-FFQ is an FFQ containing 218 food items based on the identification of IBD-specific foods [7]. Nevertheless, FFQs containing over 200 food items are more effective at ranking individuals according to their intake compared to shorter FFQs [37]. Besides, the number of questions may vary according to the purpose and the population of the FFQ. Neelakantan et al. observed that about 163 food items are sufficient to cover 95% of the energy intake consumed [38]. Additionally, it can be noted that the GINQ-FFQ was administered digitally, allowing respondents to pause and resume completion at a later time. This not only contributes to improved compliance but also positively impacts the accuracy of the FFQ responses. Therefore, the GINQ-FFQ includes a comprehensive representation of the diet of patients with IBD [7].

We observed no significant mean difference in total energy (kcal) intake between the GINQ-FFQ and 3FD. However, while agreement in energy intake between the two methods seems fair, we did observe a slightly increased difference in energy intake between the GINQ-FFQ and 3FD, as the mean energy intake of both methods increased. Furthermore, when analysing the Bland–Altman plot, patients with a low total energy intake tend to report a higher energy intake in the 3FD compared to the GINQ-FFQ, whereas patients with a higher total energy intake tend to report a lower intake in the 3FD compared to the GINQ-FFQ. After stratifying for sex, we specifically observed this for males, but not for females. These findings are consistent with other validation studies comparing food-frequency questionnaires with reference methods in healthy individuals [33,39]. Sex-specific reporting patterns may be explained by traditional health notions, where men could overreport high-calorie foods in food diaries, viewing them as better foods [40]. Furthermore, men usually have higher total energy intake due to physiological differences [41].

### 4.4. Strengths and Limitations

The IBD population tends to be rather heterogeneous, with a variety of different phenotypes and disease behaviours [1]. This validation study is based on a representative population sample of patients with IBD, due to the availability of the 1000IBD cohort [9]. With a sample size of 142 participants, the sample meets recommendations of what is known in the literature [18]. Another strength is that in both FFQ and 3FD, we calculated the intake of nutrients from nutritional supplements. This is important since nutritional studies previously have failed to include supplement use even though people increasingly use supplements [42]. This is especially the case for patients with IBD, since there is an increase in hypotheses and evidence for the benefit of the use of certain nutritional supplements as supporting treatment of inflammation as well as treatment of nutrient deficiencies [43,44]

The use of 3-day food records from the 1000IBD cohort was chosen over multiple 24 h recalls for practical reasons and to avoid unnecessarily burdening the participants with additional questionnaires, since the cohort does not have the primary objective of validating an FFQ. In choosing for 3FDs over 24 h recalls, there were no interviews, which allowed for more flexibility of time for the participants, which was more suitable for the used cohort. Furthermore, 3FDs are known to be less prone to recall bias [45]. A consequential limitation is that the present study does not account for the de-attenuation of correlation coefficients. The 3FDs were completed over a relatively short period (3 days, including 2 non-consecutive weekdays and 1 weekend day) and were not repeated throughout the year or seasons. Additionally, we were not able to use biomarkers. Accounting for de-attenuation would correct for measurement errors that influence the variation in diet. While the degree of variation differs greatly depending on the nutrient, substantial day-to-day variation in nutrient intake has consistently been observed among individuals and is inherent in the use of dietary assessment tools [46]. These variations in diet lead to measurement errors, which tend to attenuate the observed correlations in nutrients between two dietary assessment methods. Omitting de-attenuation could have resulted in an underestimation of the correlation coefficients in this validation study. Specifically, the intake of mono-/disaccharides, vitamin C, vitamin A, and cholesterol is prone to having a large within-person variation [46,47].

Ideally, the use of a biomarker to validate the nutrient intake of the GINQ-FFQ would have been appropriate. However, the serum blood and urine samples used from the 1000IBD cohort in most cases were not collected at the time of completing the 3FD and GINQ-FFQ, which would result in an inaccurate comparison. Moreover, there are limited biomarkers available for the validation of food groups. Therefore, the validation relies on self-reported intake of the GINQ-FFQ and 3FD, of which it is known that self-reported intake yields lower correlation coefficients compared to interviewer-administered nutritional assessment methods [17].

Furthermore, reproducibility was not assessed since the GINQ-FFQ was not filled out repeatedly by participants, for the same reason that multiple 24 h recalls were not administered. Both validation based on biomarkers as well as assessment of reproducibility could be part of a follow-up study using a more suitable cohort/study design.

Of particular note was that in this validation study, cross-classification for the food groups was not always feasible. This may be explained by the standardized response options used in the GINQ-FFQ, making it difficult to classify the observed food group intake into quintiles. The use of quintiles was based on the assumption that this would lead to better discrimination of intake levels, although not frequently used in FFQ validation studies [28,48,49]. Cross-classification based on quartiles is more commonly used; however, its application did not result in improvement in this validation study.

### 4.5. Future Perspectives

The GINQ-FFQ is based on a list of food items specifically developed for IBD patients, taking into account dietary strategies to cope with IBD symptoms. This allows more specific identification of food items that are considered favourable or not and thus consumed in different quantities depending on disease activity. However, these food items are not exclusively important in IBD but play a role in other immune-mediated inflammatory diseases (IMIDs) as well [50]. Therefore, the questionnaire could also be valuable to use in nutrition research investigating other populations.

In the future, the GINQ-FFQ will be implemented in different IBD study cohorts to evaluate nutritional intake, which allows for a better comparison of dietary outcomes. Furthermore, the GINQ-FFQ will be incorporated in e-health tools, which are currently used in standard clinical practice in the Netherlands. Healthcare providers can utilize the output of this digitalized FFQ to formulate personalized dietary advice for their patients.

## 5. Conclusions

Overall, the GINQ-FFQ is a sufficiently valid food frequency questionnaire to assess nutritional intake specifically for patients with IBD. However, for males with high total energy intakes, dietary assessment could be less accurate. The validated GINQ-FFQ can be used in nutrition research and healthcare to help healthcare professionals offer evidence-based dietary guidance and improve patient outcomes.

## Figures and Tables

**Figure 1 nutrients-17-00239-f001:**
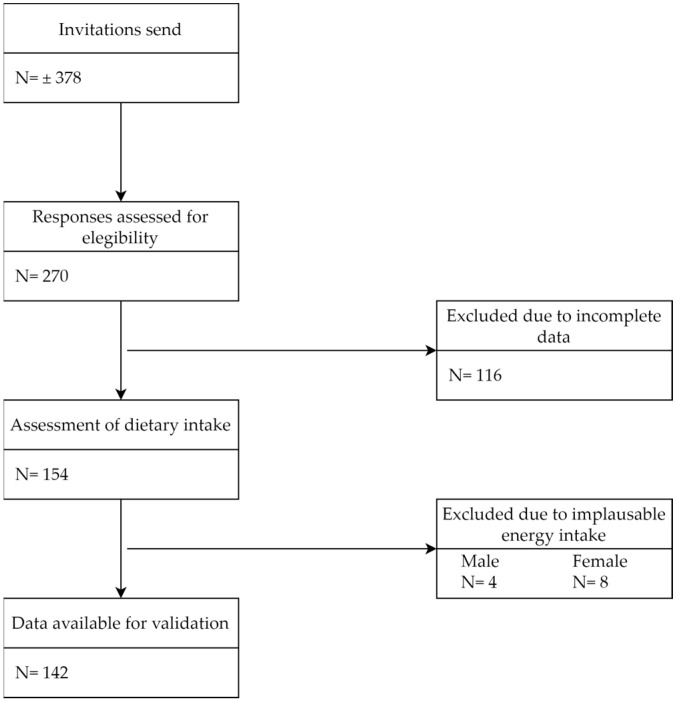
Flowchart of participant inclusion. N = number, implausible energy intake = overall daily intake for women <500 and >3500 kcal and <800 and >4000 for men.

**Figure 2 nutrients-17-00239-f002:**
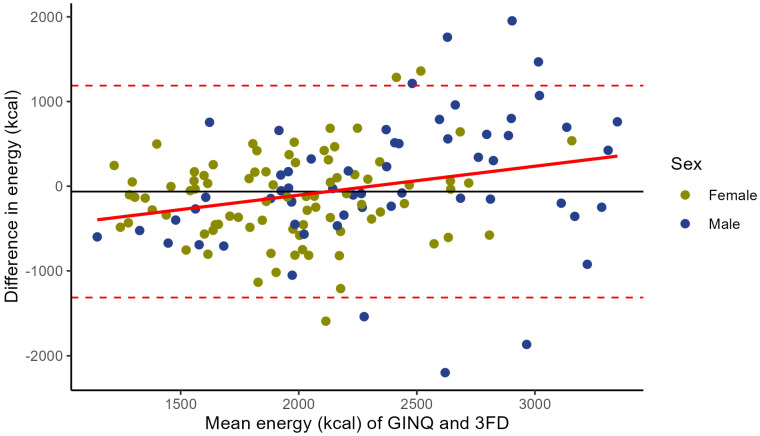
Bland–Altman plot for total energy intake based on a food frequency questionnaire (GINQ-FFQ) and a 3-day food diary (3FD) stratified by sex. Differences in total energy intake of both methods are plotted against the mean energy intake calculated by the two methods for both females and males (N = 142). The two dashed lines represent the limits of agreement (−1315, 1188), and the solid line represents the mean difference between the GINQ-FFQ and the 3FD (−63.6).

**Figure 3 nutrients-17-00239-f003:**
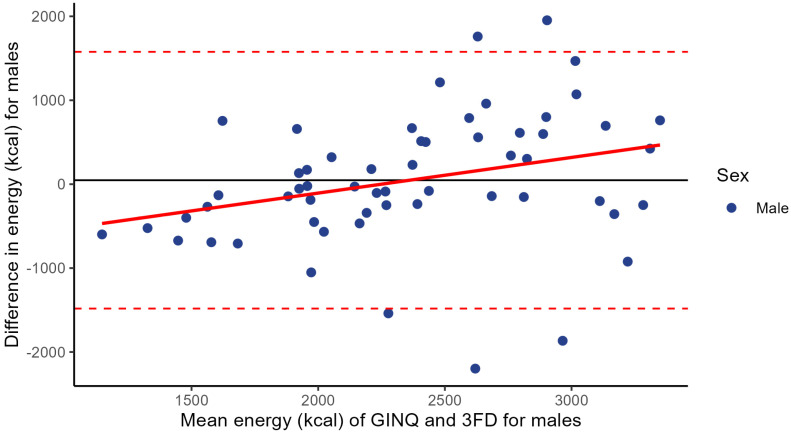
Bland–Altman plot for total energy intake based on a food frequency questionnaire (GINQ-FFQ) and a 3-day food diary (3FD) for males. Differences in total energy intake of both methods are plotted against the mean energy intake calculated by the two methods (N = 58). The two dashed lines represent the limits of agreement (−1482, 1577), and the solid line represents the mean difference between the GINQ-FFQ and the 3FD (+47.3).

**Figure 4 nutrients-17-00239-f004:**
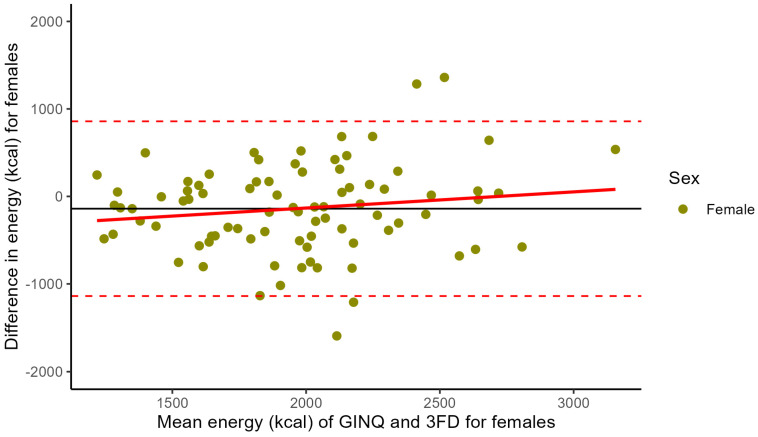
Bland–Altman plot for total energy intake based on a food frequency questionnaire (GINQ-FFQ) and a 3-day food diary (3FD) for females. Differences in total energy intake of both methods are plotted against the mean energy intake calculated by the two methods (N = 84). The two dashed lines represent the limits of agreement (−1139, 858), and the solid line represents the mean difference between the GINQ-FFQ and the 3FD (−140.2).

**Table 1 nutrients-17-00239-t001:** Patient, disease, and nutritional characteristics.

Demographics	N = 142
Female	84 (59.2)
Age (year)	49 ± 14
Weight (kg)	77.4 ± 14.3
Height (cm)	174 ± 10
Body mass index (kg/m^2^)	25.6 ± 4.5
Ethnicity	
Caucasian	139 (97.9)
Other	3 (2.1)
Smoking status	
Smoker	17 (12.0)
Previous smoker	56 (39.4)
No, never smoked	68 (47.9)
Education	
Preferred not to answer	5 (3.5)
Primary education	10 (7.0)
Secondary education	63 (44.4)
Higher education	63 (44.4)
Income (*p*/month),	
Preferred not to answer	25 (17.6)
Unknown	1 (0.7)
No income	4 (2.8)
Low income (€750–€1500)	19 (13.4)
Middle income (€1500–€3000)	46 (32.4)
High income (≥€3000)	46 (32.4)
Food-related quality of life (FrQoL)	109 ± 23
Physical activity (Baecke)	7.53 ± 1.2
Work index	2.57 ± 0.65
Sport index	2.20 ± 0.63
Spare time index	2.75 ± 0.70
**Disease Characteristics**	**N = 142**
Age at diagnosis (year)	29 ± 13
Missing	9 (6.3)
IBD phenotype	
Crohn’s disease (CD)	79 (55.6)
Ulcerative colitis (UC)	57 (40.1)
IBD-undefined (IBDU)	6 (4.2)
Monitor IBD at Home sore score (MIAH)	
CD	6.45 ± 1.48
UC and IBDU	8.38 ± 1.59
Ostomy or pouch	
Pouch	8 (5.6)
Colostomy	4 (2.8)
Ileostomy	17 (12.0)
Use of IBD specific medication	98 (69.0)
**Nutritional Characteristics**	**N = 142**
Responses to who plans/prepares meals:	
Female: Self	70 (49.3)
Female: Someone else	14 (9.9)
Male: Self	28 (19.7)
Male: Someone else	30 (21.1)
Specific diet prescriptions, dietary habits, or food preferences	41 (28.9)
Use of sip feeds	4 (2.8)
Use of tube feeds	1 (0.7)
The use of nutritional supplements (all kinds)	91 (64.1)
Type of supplement used in the previous month:	
Multivitamin use	25 (17.6)
Vitamin A	0 (0.0)
Vitamin A and D	1 (0.7)
Vitamin D	50 (35.2)
Vitamin B complex	5 (3.5)
Folic acid (vitamin B11)	6 (4.2)
Vitamin B12	21 (14.8)
Vitamin C	23 (16.2)
Vitamin E	1 (0.7)
Calcium	16 (11.3)
Magnesium	23 (16.2)
Iron	2 (1.4)
Fish oil	13 (9.2)
Other supplements	14 (9.9)

Continuous variables are presented as mean ± sd; count data as N (%). Missing data ≤ 5% not reported in this table. IBD, inflammatory bowel disease. CD, Crohn’s disease. UC, ulcerative colitis. IBDU inflammatory bowel disease unclassified. FrQoL food-related quality of life score (assessed with Fr-QoL-29; score range = 29–145; higher score reflects better FrQoL. Scores > 89.5 are good FrQoL). Baecke physical activity score (includes work, sport, and spare time index. Scored on a scale of 1–5, 5 = most activity and 1 = least activity). MIAH questionnaires for CD and UC (MIAH-CD > 3.6 = active disease; MIAH-UC > 3.5 = active disease).

**Table 2 nutrients-17-00239-t002:** Analysis of absolute nutrient intake per day from the GINQ-FFQ and 3FD: mean difference, group-level bias, cross-classification, and correlation coefficients between the two methods.

Nutrient	GINQ-FFQ (Mean ± SD)	3FD (Mean ± SD)	Mean Difference (GINQ-3FD)	(95% CI)	Paired *t*-Test *(p)*	Group-Level Bias (%)	Cross-Classification ^¥^ (%)			Correlation Coefficient (Pearson’s *r*)	
							Same	Adjacent	Extreme	*r*	*r ^†^*
Macronutrients											
Energy (kcal/day)	2092 ± 673	2155 ± 523	−63.6	(−169.5, 42.3)	0.237	−2.95	34.5	38.7	0.7	0.45 **	-
Carbohydrates											
Total (g/day)	223 ± 82	224 ± 68	−0.36	(−13.1, 12.4)	0.955	−0.16	33.1	41.6	1.4	0.49 **	0.52 **
Total (en%)	42.7 ± 6.9	41.6 ± 7.4	1.07	(−0.1, 2.2)	0.062	2.58	29.6	35.2	2.1	0.55 **	0.54 **
Mono-/disaccharides (g/day)	101 ± 45	96 ± 41	4.73	(−2.3, 11.8)	0.186	4.94	33.8	39.4	1.4	0.52 **	0.51 **
Fibres (g/day)	22 ± 8	21 ± 7	0.10	(−1.3, 1.5)	0.888	0.47	33.8	39.4	2.1	0.40 **	0.48 **
Protein											
Total (g/day)	79 ± 27	87 ± 25	−7.07	(−11.9, −2.2)	0.004 *	−8.17	29.6	43.0	2.1	0.37 **	0.41 **
Total (g/kg)	1.0 ± 0.4	1.2 ± 0.4	−0.10	(−0.2, −0.0)	0.062	−8.87	30.3	44.4	2.8	0.42 **	0.65 **
Total (en%)	15.4 ± 2.7	16.1 ± 3.0	−0.78	(−1.3, −0.3)	0.001 *	−4.86	33.8	32.4	2.8	0.49 **	0.47 **
Animal (g/day)	46.9 ± 19.8	53.0 ± 22.8	−6.08	(−9.9, −2.2)	0.002 *	−11.5	31.7	40.9	4.2	0.41 **	0.45 **
Plant-based (g/day)	32.6 ± 13.1	33.3 ± 14.3	−0.70	(−3.5, 2.1)	0.627	−2.09	31.7	39.4	4.2	0.23	0.22 *
Fat											
Total (g/day)	88.7 ± 32.6	85.8 ± 28.4	2.95	(−2.5, 8.4)	0.282	3.45	28.2	38.7	1.4	0.43 **	0.44 **
Total (en%)	38.0 ± 6.2	35.6 ± 6.9	2.36	(1.2, 3.6)	<0.001 *	6.62	22.5	38.7	2.1	0.40 **	0.39 **
MUFA (g/day)	30.6 ± 11.5	33.1 ± 12.0	−2.54	(−4.7, −0.4)	0.021 *	−7.68	23.9	44.4	2.8	0.39 **	0.35 **
PUFA (g/day)	17.1 ± 7.1	16.6 ± 6.2	0.46	(−0.8, 1.8)	0.477	2.79	32.4	35.9	2.8	0.34 **	0.22 *
Omega-3 fatty acids (g/day)	2.0 ± 0.8	2.0 ± 1.0	0.02	(−0.2, 0.2)	0.809	1.10	29.6	35.9	4.9	0.28 **	0.28 **
Omega-6 fatty acids (g/day)	14.1 ± 6.2	13.7 ± 5.5	0.49	(−0.6, 1.6)	0.396	3.56	35.2	33.1	2.1	0.33 **	0.20 *
Saturated (g/day)	33.6 ± 14.2	34.2 ± 12.2	−0.57	(−2.8, 1.6)	0.606	−1.67	34.5	34.5	0.7	0.51 **	0.60 **
Cholesterol (mg/day)	240 ± 106	234 ± 126	5.48	(−12, 23)	0.546	2.34	40.1	32.4	1.4	0.58 **	0.64 **
Alcohol (g/day)	4.5 ± 7.0	4.4 ± 8.5	0.08	(−1.0, 1.1)	0.874	1.90	-	-	-	0.69 **	0.69 **
Micronutrients											
Vitamins											
RAE (µg/day)	784 ± 421	772 ± 462	12.7	(−72, 97)	0.767	1.64	33.1	31.0	6.3	0.34 **	0.30 **
Folate (B11 natural, µg/day)	249 ± 124	268 ± 83	−19.2	(−42, 4)	0.100	−7.17	28.2	32.4	4.2	0.15	0.11
Thiamine (B1, mg/day)	1.0 ± 0.4	1.2 ± 0.9	−0.24	(−0.4, −0.1)	0.606	−20.1	28.2	33.8	2.8	0.13	0.08
Riboflavin (B2, mg/day)	1.4 ± 0.6	1.7 ± 0.9	−0.25	(−0.4, −0.1)	0.004 *	−15.3	26.1	36.6	4.9	0.16	0.12
Pyridoxin (B6, mg/day)	1.7 ± 0.7	1.7 ± 1.0	0.02	(−0.2, 0.2)	0.805	1.34	24.7	33.8	3.5	0.15	0.10
Cobalamin (B12, µg/day)	4.3 ± 2.0	4.8 ± 2.9	−0.47	(−1.0, 0.0)	0.056	−9.89	28.2	40.1	5.6	0.33 **	0.33 **
Ascorbic acid (C, mg/day)	87 ± 48	113 ± 91	−25.4	(−40, −10)	0.001 *	−22.5	24.7	43.0	2.8	0.25 *	0.27 *
Vitamin D (µg/day)	3.8 ± 1.8	4.0 ± 4.4	−0.18	(−0.9, 0.6)	0.636	−4.56	23.2	39.4	4.2	0.12	0.05
Vitamin E (mg/day)	14.0 ± 6.0	14.0 ± 6.3	−0.04	(−1.3, 1.2)	0.949	−0.29	26.8	36.6	2.1	0.26 *	0.09
Calcium (mg/day)	1020 ± 465	1076 ± 364	−56.2	(−137, 25)	0.173	−5.22	31.0	32.4	3.5	0.33 **	0.33 **
Magnesium (mg/day)	343 ± 121	345 ± 107	−2.29	(−22, 17)	0.816	−0.67	35.2	33.1	0.7	0.48 **	0.48 **
Potassium (mg/day)	3018 ± 966	3300 ± 857	−281	(−453, −110)	0.001 *	−8.52	28.9	34.5	2.8	0.36 **	0.43 **
Iron, total (mg/day)	10.3 ± 3.3	10.9 ± 3.6	−0.61	(−1.2, 0.0)	0.052	−5.60	31.0	35.9	2.1	0.42 **	0.37 **
Selenium (µg/day)	49 ± 16	50 ± 26	−0.41	(−4.4, 3.6)	0.838	−0.83	32.4	35.9	3.5	0.41 **	0.33 **
Zinc (mg/day)	11.6 ± 17.8	11.5 ± 4.2	0.07	(−2.9, 3.1)	0.963	0.62	31.7	32.4	3.5	0.06	−0.01

* *p* < 0.05, ** *p* < 0.001. Group-level bias = (GINQ-FFQ/3FD) × 100 − 100. Abbreviations: MUFA = monounsaturated fatty acid. PUFA = polyunsaturated fatty acids. RAE = retinol activity equivalents. ^¥^ Cross classification: based on quintiles. ^†^ Energy-adjusted correlation coefficients using the residual method.

**Table 3 nutrients-17-00239-t003:** Analysis of absolute food group intake in grams per day from the GINQ-FFQ and 3FD: median habitual intake, cross-classification, and correlation coefficients.

						Cross-Classification ^¥^ (%)			Correlation Coefficient (Spearman Rank)	
Food Group	GINQ-FFQ		3FD		Wilcoxon Signed Rank Test *(p)*	Same	Adjacent	Extreme	*rho* ^†^	*rho* ^‡^
	Median [IQR]	min-max	Median [IQR]	(min-max)						
Breakfast products	0 [0, 0]	0−450	0 [0, 0]	0−0	0.005 *	-	-	-	-	-
Breakfast grains	0 [0, 8]	0−82	0 [0, 10]	0−250	0.311	-	-	-	0.76 **	1.00
Dairy	250 [150, 421]	7−150	303 [205, 460]	0−1053	0.028 *	37.32	42.25	2.11	0.57 **	0.55 **
Bread and bread substitutes	111 [71, 160]	0−627	111 [74, 163]	0−538	0.500	44.37	36.62	0.00	0.69 **	0.60 **
Fats/oils	22 [12, 34]	0−92	17 [8, 27]	0−58	<0.001 **	40.14	32.39	0.70	0.51 **	0.41 **
Spreads	13 [4, 30]	0−105	16 [5, 30]	0−108	0.681	33.80	42.25	0.00	0.59 **	0.36 **
Eggs	15 [7, 29]	0−100	0 [0, 25]	0−100	0.006 **	-	-	-	0.49 **	0.54 **
Fruits	112 [56, 216]	0−1095	133 [77, 194]	0−557	0.151	40.85	30.28	0.70	0.59 **	0.45 **
Nuts/stone fruits/seeds	8 [1.7, 19]	0−109	1 [0, 15]	0−108	<0.001 **	-	-	-	0.53 **	0.28 **
Meat/meat substitutes	101 [62, 137]	0−433	93 [53, 140]	0−441	0.974	38.73	33.80	0.70	0.50 **	0.43 **
Fish	11 [5, 19]	0−71	0 [0, 16.7]	0−125	0.007 *	-	-	-	0.38 **	0.13
Vegetables	98 [60, 148]	0−592	116 [67, 192]	0−715	<0.001 **	30.99	34.51	2.11	0.34 **	0.32 **
Legumes	16 [6, 30]	0−121	0 [0, 33]	0−150	0.008 *	-	-	-	0.20 *	0.21
Grain products	99 [69, 144]	0−325	83 [33, 133]	0−387	0.002 *	27.46	42.26	2.81	0.43 **	0.35 **
Sauces	11 [6, 17]	0−96	13 [2, 31]	0−133	0.063	28.37	36.17	2.84	0.37 **	0.16
Additions to meals	1 [1, 2]	0−18	2 [0, 8]	0−81	<0.001 **	-	-	-	0.23 *	0.24 *
Fast food	23 [11, 47]	0−323	38 [0, 93]	0−293	0.005 **	-	-	-	0.27 *	0.24 *
Soup	32 [20, 41]	0−196	0 [0, 96]	0−300	0.628	-	-	-	0.32 **	0.17
Savory snacks	17 [6, 39]	0−156	8 [0, 33]	0−85	0.002 *	-	-	-	0.40 **	0.36 **
Sweet snacks	57 [28, 95]	3−826	56 [34, 85]	0−391	0.328	23.24	44.37	2.82	0.39 **	0.31 **
Beverages	1624 [1295, 2045]	715−4084	1917 [1521, 2348]	242−4782	<0.001 **	35.92	36.62	3.52	0.49 **	0.49 **

* *p* < 0.05, ** *p* < 0.001. - = not applicable. ^¥^ Cross classification: based on quintiles. ^†^ Based on log-transformed values. ^‡^ Energy-adjusted correlation coefficients using the residual method.

## Data Availability

The data presented in this study are available on reasonable request to the corresponding author. The data are not publicly available due to patient privacy legislation.

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
