# Peer review of "Relative Validity of the Groningen IBD Nutritional Questionnaire (GINQ-FFQ): A Food Frequency Questionnaire Designed to Assess Nutritional Intake in Patients with Inflammatory Bowel Disease"

_nutrients, 2025, doi:10.3390/nu17020239_

Round 1

Reviewer 1 Report

Comments and Suggestions for Authors

Dear Authors,

Congratulations on the well-structured validation of your GINQ-FFQ and the honest analysis provided. We have a few concerns and hope that our observations will help further improve an already excellent article.

Lines 74 to 78: Please provide details about the language used and, if applicable, references for the translation validation process for Dutch. Why did you choose the tools mentioned, and were they effective for the purpose of this validation study? Where in the text did you address this?  Can you clarify this and consider removing any unnecessary information? If retained, it would be helpful to provide additional details to enable future readers to interpret the results. For instance, in Table 1, how should the FrQoL results or the Baecke results be interpreted?

Lines 109 to 111: Please include a reference for this procedure and provide additional information to justify how the portion weights were estimated and with which instruments. As you know, accurately estimating portion weights is essential for reliable food diary data. Why not consider using written diaries accompanied by photographs for better accuracy?

In light of your results, particularly regarding the specific IBD food groups, do you believe the FFQ has the potential to assist healthcare providers in formulating specific dietary advice? We think this discussion is crucial, as it aligns with the primary purpose of your tool.

Author Response

Response to reviewer 1:

Comment 1: Lines 74 to 78: Please provide details about the language used and, if applicable, references for the translation validation process for Dutch. Why did you choose the tools mentioned, and were they effective for the purpose of this validation study? Where in the text did you address this?  Can you clarify this and consider removing any unnecessary information? If retained, it would be helpful to provide additional details to enable future readers to interpret the results. For instance, in Table 1, how should the FrQoL results or the Baecke results be interpreted?

Response 1: Thank you for your remark, we indeed mention questionnaires filled out as part of the 1000IBD cohort, that were not used for validation of the GINQ-FFQ. The questionnaires that were used, are part of assessing baseline and disease characteristics of our validation cohort. Therefore, we adapted this part of the method section “Apart from the GINQ-FFQ, participants additionally filled out questionnaires in Dutch to assess baseline characteristics of the 1000IBD cohort through the REDCap data capturing tool. These included questionnaires to evaluate disease activity (Monitor IBD At Home questionnaire (MIAH)) [12], food-related quality of life (Fr-QoL-29) [13], physical activity (Baecke questionnaire) [14] and some questions to assess socio-economic status and educational level.” (line 73-78). We clarified that the language used is Dutch (line 74). Since FrQoL is not part of the GINQ-FFQ, but only the assessment of baseline characteristics, we did not discuss the validation of the FrQoL in the method section. The FrQoL was developed specifically for the IBD population. We translated all questions directly from English to Dutch. Our supervisor has been in contact with the developer of the FrQoL questionnaire, Kevin Whelan. For all presented questionnaires, such as the Baecke, ranges to use for interpretation are already described in the footnote of the table “FrQoL food related quality of life score (assessed with Fr-QoL-29; score range= 29-145; higher score reflects better FrQoL. Scores >89.5 are good FrQoL). Baecke Physical activity score (includes work, sport and spare time index. Scored on a scale of 1-5. 5= most activity, 1= least activity.). MIAH questionnaires for CD and UC (MIAH-CD >3.6= active disease; MIAH-UC >3.5= active disease).”. We opted to not further describe these in the text.

Comment 2: Lines 109 to 111: Please include a reference for this procedure and provide additional information to justify how the portion weights were estimated and with which instruments. As you know, accurately estimating portion weights is essential for reliable food diary data. Why not consider using written diaries accompanied by photographs for better accuracy?

Response 2: Thank you for pointing out this omission. Participants weighted/measured this by themselves. Instructions were put clearly on the paper 3-day-food diary (3FD) they received. To clarify, we added an extra supplement showing an example of this 3FD and the instruction the participants received  “Supplement 2 shows the English translation of the used 3FD with all the instruction participants received.” (S2, line 115-116). Furthermore, we stated self-reporting of the food intake more clearly by adding  “The collection of this data was fully self-reported, without direct supervision of a healthcare provider.” (line 110-111).

Comment 3: In light of your results, particularly regarding the specific IBD food groups, do you believe the FFQ has the potential to assist healthcare providers in formulating specific dietary advice? We think this discussion is crucial, as it aligns with the primary purpose of your tool.

Response 3: Thank you for your remark. We have added an extra section to our discussion to elaborate on purpose of the GINQ-FFQ in a paragraph on future perspectives

“4.5. Future perspectives

The GINQ-FFQ is based on a list of food items specifically developed for IBD patients, taking into account dietary strategies to cope with IBD symptoms. This allows more specific identification of food items, that are considered favourable or not, and thus consumed in different quantities depending on disease activity. However, these food items are not exclusively important in IBD, but play a role in other immune-mediated inflammatory diseases (IMIDs) as well [51]. Therefore the questionnaire could also be valuable to use in nutrition research investigating other populations. 

In the future, the GINQ-FFQ will be implemented in different IBD study cohorts to evaluate nutritional intake, which allows for a better comparison of dietary outcomes. Furthermore, the GINQ-FFQ will be incorporated in e-health tools which are currently used in standard clinical practice in the Netherlands. Healthcare providers can utilise the output of this digitalised FFQ to formulate personalised dietary advice for their patients. ” (line 445-456).

Furthermore, we have also added an extra sentence to our conclusion as suggested by another reviewer “The validated GINQ-FFQ can be used in nutrition research and healthcare to help healthcare professionals offer evidence-based dietary guidance and improve patient outcomes.” (line 463-465).

Reviewer 2 Report

Comments and Suggestions for Authors

This manuscript examines the relative validity of a food frequency questionnaire to assess nutritional intake in comparison with a 3-day food diary, specifically in patients with Inflammatory Bowel Disease. In my opinion, authors have carried out a comprehensive work, but I would like to add some suggestions:

-          Figure 1 could be improved to increase legibility and make it more attractive.

-          Line 106: “For the 3FD, patients were asked to write down their nutritional intake…” Overall, to collect this kind of data is broadly recommended to be carried out by specialist face-to-face interviewers. This point has been properly included as a limitation, but please, specify that questionnaires are self-reported (it must be clear if there were no supervision) in the “method” section.

-          It is difficult to understand how a FFQ (where you use to just specify food consumption in times per day/week/month) can be converted into the amount of nutrients consumption.  Could you detail or provide more information about how the FFQ Tool™ works? I imagine more variables must be taken into consideration (e.g. portion size).

-          Maybe the “conclusion” section is a little bit short. You could add some sentences highlighting the possible applications/advantages of this tool.

Author Response

Response to reviewer 2:

Comment 1: Figure 1 could be improved to increase legibility and make it more attractive

Response 1: Thank you for your remark. After re-evaluation of the figure (figure 1, line 95), we have changed the fond of the letters and numbers and colour of the boxes to resemble our tables. This also makes the text in the figure more readable. 

Comment 2: Line 106: “For the 3FD, patients were asked to write down their nutritional intake…” Overall, to collect this kind of data is broadly recommended to be carried out by specialist face-to-face interviewers. This point has been properly included as a limitation, but please, specify that questionnaires are self-reported (it must be clear if there were no supervision) in the “method” section.

Response 2: Thank you for this comment. We have now stated this more clearly “The collection of this data was fully self-reported, without direct supervision of a healthcare provider.” (line 110-111) . We have also added the 3FD with all the instructions to the participants, as a supplement and added “Supplement 2 shows the English translation of the used 3FD with all the instruction participants received.” (S2, line 115-116).

Comment 3: It is difficult to understand how a FFQ (where you use to just specify food consumption in times per day/week/month) can be converted into the amount of nutrients consumption.  Could you detail or provide more information about how the FFQ Tool™ works? I imagine more variables must be taken into consideration (e.g. portion size).

Response 3: Thank you for this remark. The way the FFQ tool works is very elaborate to describe, this is too be found fully in the referenced thesis (reference 24). However, to ensure readers of this manuscript to understand how the calculation is made, we added description of the most important elements in our method section “The FFQ-ToolTM uses ranges of standard household measures and common Dutch portion sizes and frequency questions with specific weightings to determine the daily amounts used on average over the past month [24]. To convert the daily intake amounts into nutrient intake, the FFQ-ToolTM was linked to the Dutch food composition database (NEVO, version 2010/2.0) [25].”  (line104-108)

Comment 4: Maybe the “conclusion” section is a little bit short. You could add some sentences highlighting the possible applications/advantages of this tool.

Response 4: Thank you for your suggestion. We have expanded the conclusion paragraph “The validated GINQ-FFQ can be used in nutrition research and healthcare to help healthcare professionals offer evidence-based dietary guidance and improve patient outcomes.” (line 463-465). Furthermore, based on suggestion by another reviewer, we have also added a small section to our discussion

“4.5. Future perspectives

The GINQ-FFQ is based on a list of food items specifically developed for IBD patients, taking into account dietary strategies to cope with IBD symptoms. This allows more specific identification of food items, that are considered favourable or not, and thus consumed in different quantities depending on disease activity. However, these food items are not exclusively important in IBD, but play a role in other immune-mediated inflammatory diseases (IMIDs) as well [51]. Therefore the questionnaire could also be valuable to use in nutrition research investigating other populations. 

In the future, the GINQ-FFQ will be implemented in different IBD study cohorts to evaluate nutritional intake, which allows for a better comparison of dietary outcomes. Furthermore, the GINQ-FFQ will be incorporated in e-health tools which are currently used in standard clinical practice in the Netherlands. Healthcare providers can utilise the output of this digitalised FFQ to formulate personalised dietary advice for their patients. ” (line 445-456).